# Colour Preference and Healing in Digital Roaming Landscape: A Case Study of Mental Subhealth Populations

**DOI:** 10.3390/ijerph191710986

**Published:** 2022-09-02

**Authors:** Tao Huang, Shihao Zhou, Xinyi Chen, Zhengsong Lin, Feng Gan

**Affiliations:** 1Virtual Landscape Design Laboratory, School of Art and Design, Wuhan Institute of Technology, Wuhan 430205, China; 2School of Foreign Languages, Wuhan Institute of Technology, Wuhan 430205, China; 3Tus-Design Group Co., Ltd., Suzhou 215000, China; 4School of Art, Culture and Tourism Industry Think Tank Chinese Art Evaluation Institute, Southeast University, Nanjing 211189, China

**Keywords:** emotional preference, landscape colour, healing effect, mental subhealth

## Abstract

From the perspective of emotional preference, studies have been conducted about landscape healing for subhealth people in the National High-Tech Zone (the study area). Combined with a preliminary practice investigation, Unity 2019 was used to make a digital roaming landscape, and 91 subjects with a history of mental subhealth diseases were randomly invited to participate in the immersive experimentation of the Human–Machine Environment Synchronisation (ErgoLAB) platform. After the experiment, the subjects were invited to fill in a Likert scale as the control group. The interest preference, emotion fluctuation, and healing effect of landscape colour were verified. The results show that: (1) The variation trend of interest reflected the concentration of interest in landscape, and the order of degree of interest, ranging from high to low, went Green > Yellow > Blue > Red > Orange > Purple > Cyan. (2) The subjects’ interest in landscape colour was correlated with the arousal of positive emotions. The correlation between interest in landscape colour and positive emotions, from high to low, went Blue > Green > Yellow, while the correlation between red, cyan, orange, and purple was not significant. (3) The mean skin conductance (SC) fluctuation variance of subjects was 5.594%, which confirmed that the healing effect of digital roaming landscape scenes was significant under the state of low arousal. According to the Likert scale data, subjects’ scores of the healing effect of landscapes, from high to low, went as follows: Green > Yellow > Red > Blue > Cyan > Orange > Purple. The results provide a new method for demonstrating the logical relationship between the digital landscape interest experience–emotional awakening–healing effect and providing a theoretical method and construction scheme for landscape colour configuration in the implementation of landscape healing projects.

## 1. Introduction

As COVID-19 continues to wreak havoc, high-pressure, fast-paced work and lifestyles have led to a series of noteworthy social problems, such as mental subhealth, chronic disease, and a younger age for potential disease to emerge. According to an investigation by the World Health Organisation (WHO), 5% of people worldwide are healthy, 20% are sick, and 75% are in subhealth, which seriously affects the quality of life and work [1]. The statistical results of the subhealth rating scale from six provinces in China show that the overall incidence of subhealth in urban residents is 68.06%, the incidence of mental subhealth is 65.96%, and the incidence of social subhealth is 70.76% [2]. Approximately 200–300 million people in China have been found to be suffering from mental problems, including depression. Anxiety is the most prevalent mental problem [3]. Helping those with psychological subhealth status relieve anxiety and tension caused by life and work pressures and reducing the transition from mental subhealth to physical illness, mental illness, and even mental crisis have become significant problems that call for urgent solutions.

Traditional Chinese medicine (TCM) believes that man and nature are inseparable as a unity, and this relationship is the most fundamental one in the world [4]. The theory of landscape ecology argues that people’s physical and psychological perception should be included in ecological environment research. Physical perception refers to people’s perception of their scene through their senses, while psychological cognition refers to people’s psychological reaction formed through physical perception [5]. Colour psychology focuses on the relationship between the phenomenon of colour composition and human psychological consciousness. Through people’s visual colour, it exerts an influence on perception, emotion, memory, and other emotions, among which the colour of the living environment is closely related to the development of human psychology [6,7,8]. Shen et al. (2021) and Dupont et al. (2017) argued that in a landscape with coordinated colour collocation, people can directly stimulate nerves by feeling the colours of different landscapes, which plays a role in conveying emotion, relaxing the body and mind, and fostering interpersonal communication, and thus has a positive impact on people’s physiological, mental, and emotional health [9,10]. However, most studies in this area focus on the colour function design collocation principle without exploring the construction of roaming landscape scenes and immersive experiences, and thus, the evaluation of the final result is uncertain. Kelly and Heit. (2017), L.C. et al. (2010), and Rodiek and Fried. (2005) used qualitative and quantitative research methods, such as the dual comparison method, interview method, and physical analysis method, to investigate objects’ visual perception and colour preference [11,12,13]. Lipson-Smith et al. (2020) and Díaz-Barrancas et al. (2020) studied the visual elements of colour in space through virtual reality (VR) technology, carried out spatial construction and immersive colour presentation, invited subjects to conduct eye movement experiments, and completed the effectiveness and reliability of the evidence-based design process [14,15]. However, no related experiments in skin electrography were conducted, and the analysis of the explicit emotional processes of subjects was insufficient [16].

Attention restoration theory shows that attention can be directed or nondirected, and directed attention plays an important role in human health. It has been found that watching more natural landscapes can strengthen people’s directed attention, thus relieving stress and healing the body and mind [17,18]. Psychological decompression theory shows that the natural environment can restore the nervous system, reduce stress, and inhibit automatic arousal. The sentiment preference rating method can be used to study psychological decompression. It was found that plants with green crowns could evoke more positive emotions than those with other colours [19,20,21,22]. Through a questionnaire survey, Küller et al. (2006) found that colour has a strong impact on people’s emotions, but the two research methods are subjective and lack support from a comparison of objective experimental data [23]. Lee and Park (2017), Novak and Richardson (2012), and Qi et al. (2017) applied “colour therapy” research to show that a certain colour can be used for spatial configuration to cure various mental or physical diseases, but most studies are on the colour configuration of indoor spaces [24,25,26,27], such as nursing homes and hospitals, and few on the healing colour configuration of outdoor spaces, such as landscapes. Wiederhold (2004) used VR technology to compare patients with anxiety in the virtual reality treatment group and other treatment groups, and the results showed that the virtual reality treatment group had a significantly better healing effect than other control groups [28], among which physiological measurement results also significantly verified the positive transformation of emotions [29]. Although the application of VR has gradually become more popular [30,31,32], a complete evaluation model of the healing effect of VR has not yet been formed in landscape therapy research under colour preference for people in mental subhealth states. The experimental scenes constructed by virtual reality technology have better performance in terms of immersion, realism, third dimension, depth of field, and even comfort [33,34,35]. In the existing research, most of the observation, questionnaire, interview, and other methods of investigation focus on the subjective behaviour and feelings of the subjects and use it as an evaluation criterion for environmental health benefits, while there are fewer related studies on the acquisition of objective parameters such as virtual environment experiments as agent indicators. Virtual reality is a feasible method for studying the colour of the situation, and the environment that presents the colour can affect the mood and valence response [36].

The literature supports that landscape plays a significant role in healing, and that colour plays an important role in landscape. However, from the perspective of theoretical research, although there are studies on colour psychology, landsenses ecology, attention restoration theory, and psychological decompression theory, there are few studies on landscape healing under the colour preferences of people in mental subhealth states in China, and no systematic theoretical framework system or design scheme has been established. Although research at the intersection of digital roaming landscape and healing has made some progress, it is still at the exploratory phase, and colour configuration is rarely involved as a focal theme in the establishment of landscape healing exploration systems and for the evaluation of healing effects, which needs further exploration. Regarding research methods, most existing studies adopt questionnaire surveys, interview methods, and object analysis methods to obtain qualitative or quantitative research results, but there is no comprehensive comparison of objective experimental data as a support. With the rise of the Metaverse and the popularisation of VR, healing research can introduce this technology in indoor environments such as health facilities and institutions. Furthermore, colour preference and its healing relationship with the landscape can be used to explore crowd interest and the emotional healing effects; however, there is no multimethod iterative method, such as combining with electrical skin experiments, to explore the healing effect. From the perspective of policy mechanisms, although China has proposed the 14th five-year digital economy construction policy, in terms of policy implementation, the local digital infrastructure has yet to be established and improved, and the level of digital application of public services is low. Hence, digital transformation still has a long way to go.

In summary, based on the prominent problem of people with submental health, this paper first proposes an innovative model of landscape roaming design, immersive experience, emotion of interest, and healing effect testing. Second, to verify the science and rationality of the model, the objective experimental physiological indicators and subjective evaluation indicators were verified to obtain data on emotion and the healing effect of landscape colours on people with mental subhealth. The application value of this study is to heal the landscape design for actual construction enterprise design sectors to propose effective measures to alleviate the anxiety and stress caused by the national population’s mental subhealth problems, reduce the transition from mental subhealth state to physical disease, mental disease, and even psychological crisis, and provide new thoughts for the evaluation of the healing effect of landscapes.

## 2. Materials and Methods

### 2.1. Overview of Research Area

Located in southeastern Wuhan city, the East Lake High-tech Zone is the National Opto-electronics Industry Base and Pilot Free Trade Zone of China, with a planned area of 518 square kilometres that encompasses a number of universities, colleges, and provincial and ministerial research institutes. The Baozixi Ecological Corridor is the core area of the East Lake High-tech Zone, located south of Gaoxin Avenue, east of Optics Valley 4th Road, west of Baoxi Road, and north of Gaoxin 5th Road (30°45′48″ N-111°49′42″ E), with an area of approximately 1199 hectares and length of 5.09 km, as shown in Figure 1.

The people working and residing in this area encompass a wide range of professional backgrounds, including academics, researchers, experts, teachers, students, and others. However, as the pandemic situation has escalated, the high-pressure and fast-paced work and lifestyles, anxiety, depression, tension, and other negative emotions have exacerbated, and people with mental subhealth are transitioning to physical subhealth and increased risk of disease. Mz et al. (2022) define an ecological corridor as a public space for people to interact, admire, entertain, rest, and partake in other activities, and its function also undergoes constant change [37]. In addition to serving basic ecological and spatial landscape functions, ecological corridors can also improve and regulate users’ physical and mental health. Plants serve as an important part of the ecological corridor landscape and serve functions such as viewing, relieving tension, calming, and absorbing dust. At present, the landscape area of flower vegetation patches in the ecological corridor is unevenly distributed and has limited varieties. The colour collocation of plants and the landscape needs of people have not been considered, and the aesthetic and healing effects of the scene are insufficient. Thus, it is urgent to reform and redesign the ecological corridor. A well-planned and exquisitely designed ecological corridor can encourage residents and visitors to slow down and relax and even improve the healing effect of the whole ecological corridor. Therefore, constructing a healing roaming landscape in the prime areas of the urban core district is particularly important. In this paper, VR technology and electrodermal activities (EDA) experiments were introduced. In the experimental design, subjects were randomly invited to participate in the roaming landscape experience, and the emotional awakening of interest preference and the healing feedback effect of landscape colours on people in mental subhealth were measured through “human–computer–environment” interaction.

To study the emotional arousal of interest preference and the healing feedback effect under colour preference in people with mental subhealth, a practice investigation, a VR eye-tracking experiment, an EDA experiment, interviews, and a Likert scale were combined to carry out the study, and we propose the following hypotheses: (1) the participants’ interest in the colours of landscapes in the roaming landscape correlates with their emotional arousal; (2) the change in emotional arousal is positively correlated with the healing effect of landscape colour preference; and (3) the combination of subjective evaluation and physiological indicators is more objective and scientific than other methods.

### 2.2. Data Collection

Colour psychology theory was established through a literature review. Based on field investigation, terrain, vegetation, and building data in the study area were summarised by using DJI Inspire UAV II, a rangefinder, and a TITAN360 panoramic camera. Combined with field and online questionnaire surveys, landscape colour and related colour landscape category data were obtained from mental subhealth populations. At the same time, we conducted interviews with psychologists in relevant hospitals in different provinces and cities through online conferencing, telephone consultation, etc., to draw relevant suggestions.

First, the questionnaire obtained demographic information of the subjects, including gender, age, and occupation, whether they had symptoms of subhealth and other related conditions, and whether they had treatment needs. Second, the questionnaire included a scale of positive and negative emotion adjectives of landscape colour and related colour plant names preferred by the crowd. The participants were asked to: associate monochrome (red, orange, yellow, green, cyan, blue, and purple) plants in the daily landscape scenes; combine related emotional adjectives (positive emotional adjectives (PA)): active, enthusiastic, happy, elated, excited, proud, glad, energetic, vibrant, and interested, and negative emotional adjectives (NA): restless, sad, afraid, nervous, scared, guilty, irritable, discreet, angry, upset, and assign the corresponding value to match the emotion (Much more—5 points, Many—4 points, Medium—3 points, Some—2 points, Little or none—1 point); and fill in the name of the preferred colour-related landscape category in daily landscape scenes. Then, the participants were asked about their preference for complementary contrasting colours (red, green, yellow, purple and blue, orange) and cold and warm colours (blue, green, purple and red, orange and yellow) in plants.

### 2.3. Scene Construction

In field surveys as well as online surveys, 207 subjects participated in the survey, 139 of whom showed that they had symptoms of mental subhealth, accounting for 67.15% of respondents, which indicates that the mental subhealth problem within the population is serious. SPSS 21.0 software was used to test the reliability and validity of the subjective questionnaire results. The results show that Cronbach’s coefficient (*α*) = 0.873, indicating that the reliability is reliable. The principal component analysis of the questionnaire results indicated that *KMO* = 0.851, which confirmed that the results are statistically significant (*p* < 0.01) and have good structural validity. The results of the questionnaire scale show (Table 1) that the positive emotions associated with red, yellow, green, and blue plants are stronger among people in mental subhealth, while the effect of orange, purple, and cyan plants are weaker. The negative emotion associated with purple, cyan, and orange plants are also strong, while red, blue, and yellow plants are also relatively weak, with green plants being the weakest. In the complementary contrasting colours collocation with red and green plants collocated, there was no significant difference in the preference between cold and warm colour plants.

Based on the survey data, the landscape scenes with green, red, yellow, and blue plants accounted for a large proportion, while purple, cyan, and orange plants accounted for a small proportion. Red and green combinations accounted for a large proportion and were constructed based on the colour preferences of people in mental subhealth. The landscape category was selected from the relevant colour landscapes favoured by people in mental subhealth states according to the questionnaire, and the plants were planted with matching based on the field investigation. The landscape colour number was selected from the hue, saturation, and values (HSV) colour space. HSV space needs to be transformed through RGB space. The specific colour quantisation and coding method is shown in Formula (1), and Figure 2 is the HSV colour space coordinate system. The *H*, *S*, and *V* components represent the hue, saturation, and value of selected landscapes, respectively (Table 2). Considering the research object of this paper, the *H* (hue) component was selected for study.
(1)$$V=\max(R,G,B)\;S=\{\frac{V-\min(R,G,B)}{\;V\;}\begin{array}{c} ,V\neq 0\\ ,0 \end{array}\;S=\{\begin{array}{c} \frac{60(G-B)}{V-\min(R,G,B)},V\neq R\\ 120+\frac{60(B-R)}{V-\min(R,G,B)},V=G\\ 240+\frac{60(R-G)}{V-\min(R,G,B)},V=B \end{array}$$

The digital roaming landscape was built using a combination of audio–visual methods; the landscape (including sky, roads, vegetation, and buildings) in the scene corresponds to its visual elements, and the music setting in the unified scene corresponds to the auditory elements. First, 3D Studio Max software was used to build a virtual landscape scene, which was imported into Unity 2019 for VR-plugin program settings, and then box collider collisions and corresponding scripts were added so that the eye tracker could detect the object and record eye tracking data. Second, the EDA device was used to adjust the position of the Camera Rig helmet so that the height of the visual level was consistent with the height of the subject’s visual level. Finally, the exe procedure was exported to connect with the ErgoLAB platform. The participants had an immersive experience on the roaming platform.

### 2.4. VR Eye-Tracking Experiment and EDA Experiment

In this experiment, the colour changes of landscapes were selected as independent variables, and the subjects with/without interest, with/without emotional arousal, and with/without healing effect were selected as dependent variables. The principal of the experiment organised the subjects to become familiar with the experimental environment, procedures, and instructions to ensure the accuracy and reliability of the experiment. In the front stage, all subjects repeatedly practised executing the instructions and signed informed consent, wore helmets and skin electrical equipment properly, and then carried out calibration on the ErgoLAB platform to ensure that the error of five line of sight points was no more than 20 pixels to ensure that the skin electrical fluctuation frequency was normal. The sampling rate was 120 Hz. In the experimental stage, the principal instructed the subjects to conduct virtual immersion roaming in the scene. The subjects could stop and watch in the area of interest and repeat the above steps until 91 subjects (including personnel of Xiaomi and Huawei, designers of China Construction Third Bureau, and university researchers, teachers, residents, and students) with a history of mental subhealth completed the experiment (male: 50; female: 41; naked eye ≥ 1.0. Please see annex A for specific subject information). Experimental scenes and data detection are shown in Figure 3. Finally, the experimental data are exported while observing the integrity of the experimental data. Thirteen samples with larger errors are removed, and seventy-eight valid samples are obtained, with an effective sample rate of 85.71%. In the post-test stage, subjects were invited to conduct interviews and fill in Likert scales to obtain the data of interest degree, emotional preference, and healing effect of landscape colours.

### 2.5. Analysis of Experimental Data

#### 2.5.1. Visual Analysis of Interest Experience Data

Through VR eye-tracking experiments and skin electrical detection, the logical relationship between interest experience, emotional arousal, and healing effect needs to be verified. Its working principles are as follows:

After 90 s of immersive experience in the digital roaming landscape, the experimental data were extracted through the ErgoLAB platform and normalised. To confirm that different plant colours can stimulate participants’ interest in landscape colours, the participants’ interest in landscape colours was used as a standard for measurement. The degree of interest represents the weight of interest in various landscape colours. The larger the weight is, the higher the intensity of interest in a certain kind of colour. This paper calculated the degree of interest according to the fixation time and pupil diameter of the participants in the immersive experience:(2)$$Y_{ij}=\frac{\max x_{ij}-x_{ij}}{\max x_{ij}-x_{ij}};i\in \left[1,m\right],j\in \left[1,n\right]\;\&\;Y_{ij}=\frac{x_{ij}-\min x_{ij}}{\max x_{ij}-\min x_{ij}};i\in \left[1,m\right],j\in \left[1,n\right]$$
(3)$$f_{ij}=Y_{ij}/\sum_{j=1}^{n}Y_{ij}$$
(4)$$I_{i}=\frac{t_{i}}{\sum_{i=1}^{n}t_{ij}}+\frac{c_{i}}{\sum_{i=1}^{n}c_{ij}}+\frac{p_{i}}{\sum_{i=1}^{n}p_{ij}}$$

In the formula, *I_i_* is the *i*th degree of interest, *t_ij_* is the time from the *i*th fixation point to the *j*th fixation point, *c_ij_* is the number from the *i*th fixation point to the *j*th fixation point, *p_ij_* is the pupil diameter from the *i*th fixation point to the *j*th fixation point, and *n* is the total number of fixation points.

To analyse the gathering and dispersing relations among the seven colour interest degrees, SPSS was used to generate a scatter graph matrix. As seen from Figure 4, there is a strong correlation between the interest degrees of the two colours.

#### 2.5.2. Analysis of Interest Experience and Emotional Arousal

To investigate the relationship between interest degree and emotional arousal, correlation analysis between interest degree and SC wave value was conducted to verify whether there was a significant relationship between the degree of interest and emotional arousal toward landscape colour. In combination with the interview results of the subjects after the experiment, the emotional arousal results were compared and analysed by subjective evaluation indexes and objective experimental physiological indexes to comprehensively test the emotional arousal degree of the subjects in the scene experience of landscape colours.

#### 2.5.3. Analysis of Emotional Arousal and Healing Effect

Skin resistance and conductance of the human body change with the function of skin sweat glands. These measurable skin electrical changes are called EDA [38], which are composed of SC, skin conductance level (SCL), and skin conductance response (SCR) [39,40,41]. SC is the most sensitive index to evaluate the level of emotional arousal, which can reflect the overall response and the degree of interest of people in different psychological and physiological states. When people are in a tense mood, sweat secretion increases, the resistance of trace current through sweat decreases, and SC increases; conversely, in a relaxed mood, SC levels decrease. Therefore, the resistance encountered by a small amount of electric current passing through the skin can be used to measure the emotional response of the autonomic nervous system. SC signals changed significantly with different emotions. After the experiment, participants were invited to fill in The Positive and Negative Affect Scale (PANAS) to obtain whether the participants generated positive or negative emotions after experiencing the roaming scene. Combined with EDA and Likert scale data, we analysed the effects of the landscape colour with or without mood fluctuations and healing effects. By intercepting the difference between the average rate (baseline) of 1 min 01–20 s from 20 s after stimulation and the average rate (baseline) of the first 0–10 s, the corresponding extreme deviation values of SC were calculated to analyse SC fluctuation values, as shown in Figure 5.

Ward and Marsden (2003) [42] found that when subjects browsed web pages that did not follow design principles, their emotional stress and SC fluctuations increased. When the SC wave value fluctuates by at least 7%, sympathetic activity dominates, emotional arousal is high, and psychological cognitive load increases. When subjects were surveyed to browse a web page that followed design principles, the SC fluctuation is less than 7%, indicating low-stress environmental conditions. When the SC wave value is less than 7%, the fluctuation of SC showed a downwards trend, the parasympathetic nervous system dominated, the mood returned to a calm state, and the psychological cognitive load was low. This paper used SC fluctuations to reflect the emotional state of the participants. When SC was less than 7%, the subjects were in low-stress, calm, and soothing environmental conditions, which can produce healing effects. When SC is at least 7%, the subjects are under high-stress environmental conditions characterised by emotional tension.

### 2.6. Methods and Logical Structures of Research

Based on the logical relationship between interest experience, emotional arousal, and healing effects, this study analysed the degree of interest, emotional change, and healing effect of landscape colour preference in people with mental subhealth. First, the theory of colour psychology was determined through a literature review. Based on field investigation, data were collected using equipment such as DJI Inspire II drones, rangefinders, and TITAN360 panoramic cameras, and combined with on-site questionnaire scale surveys to obtain colour preferences and related colour landscape category data of people with symptoms of psychological subhealth. Second, optimised survey data and built virtual scenes and participants with a history of psychological subhealth were invited to participate in VR eye-tracking and EDA experiments, and the ErgoLAB platform was used to record the experimental data of the participants. Finally, the experimental and interview results were combined with the data from the Likert scale for pattern testing. The process can be summarised by theory preparation, scene building, immersive experience and healing effect evaluation, and pattern testing. Figure 6 is the reasoning diagram of landscape reconstruction design and analysis of healing effects.

## 3. Results

According to the pretest field survey, mid-term immersive experience experiment, post-test interview, and Likert scale verification, the logical relationship of interest experience, emotional change, and healing effect was statistically analysed, and the following results were obtained:

### 3.1. The Trend of Change in the Degree of Interest Reflects the Concentration of Participants’ Interest in Landscape Colours

To obtain the subjects’ interest in the 3D roaming landscape scene, fixation duration, fixation frequency, and pupil diameter were used to calculate subjects’ level of interest in roaming scenes. According to Formulas (2)–(4), the changing trend of the degree of interest and the overall mean value of the 78 subjects were obtained, as shown in Figure 7 and Figure 8.

According to Figure 8, when the degree of interest of the subjects after the immersion experience was at least 1.13, the subjects had high interest in the 3D landscape roaming scenes. In landscape roaming scenes, the average interest degree of green, yellow, and blue is 3.85, which was the highest degree of interest. Red, orange, and purple garnered moderate interest, and cyan had the lowest degree of interest at 2.82. To more accurately indicate the degree of interest in landscape colour, the following analysis was made. There were 77 subjects with an interest in green and red plants of at least 1.13, accounting for 98.718% of the total number of subjects, and only one subject < 1.130, accounting for 1.282% of the total; 78 subjects with an interest degree of yellow plants ≥ 1.130, accounting for 100%; 75 subjects with an interest degree of blue plants > 1.130, accounting for 96.154% of the total, and three subjects less than 1.130, accounting for 3.846%; 72 subjects with an interest degree of orange plants ≥ 1.130, accounting for 92.308% of the total, and 6 subjects < 1.130, accounting for 7.692%; 74 subjects with an interest degree of purple plants ≥ 1.130, accounting for 94.872% of the total, and 4 subjects < 1.130, accounting for 5.128%; 66 subjects with an interest degree of orange plants ≥ 1.130, accounting for 84.615% of the total, and 12 subjects < 1.130, accounting for 15.385%. The results show that the subjects showed high interest in the colours of seven types of landscapes in the roaming landscape and confirmed that the subjects were interested in the digital roaming landscape scenes in the immersive experience, which was consistent with the research results of Wu et al. (2014) [43]. Participants were interested in landscape colours in order from high to low: Green > Yellow > Blue > Red > Orange > Purple > Cyan. The results show that green, yellow, and blue should be selected as the main colours of landscape design in landscape implementation projects, red can be appropriately applied to landscape design, and the use of cyan and purple in landscape design should be appropriately reduced to improve the interest of the research objects of the project in landscape scenes.

### 3.2. Participants’ Interest in Landscape Colours Correlated with Positive Emotional Arousal

To investigate whether the relations of the interest of landscape colours and positive emotional arousal in digital roaming landscape scenarios are significant, this paper analysed the correlation between the interest of seven types of landscape colours and SC. As shown in Table 3, SC was significantly correlated with green plants (*r* = 0.266, *p* = 0.019), yellow plants (*r* = 0.252, *p* = 0.026), and blue plants (*r* = 0.348, *p* = 0.002), and had no significant correlation with red plants (*r* = 0.170, *p* = 0.136), cyan plants (*r* = 0.183, *p* =0.109), orange plants (*r* = −0.040, *p* = 0.725), or purple plants (*r* = 0.091, *p* = 0.428), which means it had no statistical significance. The results show that the subjects’ interest in digital roaming landscape was correlated with their positive emotional arousal, the subjects’ interest in green, yellow, and blue plants was high, and their positive emotion was significant. The data derived from the ErgoLAB platform show that the mean value of SC of 78 subjects was 4.32 μS, as shown in Figure 9. The results show that the subjects presented a low-arousal positive emotional state in the digital roaming landscape scene, and the average value of SC was consistent with the research results of [44]. The subjects’ interest in landscape colours was significantly correlated with positive mood, and from high to low, it went Blue > Green > Yellow, while the correlations between Red, Cyan, Orange, and Purple were not significant; thus, there was no statistical significance. The results show that in the landscape implementation project, to enhance the positive mood of the study subjects on the landscape scene, more green, yellow, and blue plant configurations should be selected, and the red, cyan, orange, and purple configurations should be appropriately reduced. Combined with the interview results after the experiment, it was further verified that when the subjects saw the blue, green, and yellow plants during the immersive experience, their mood fluctuations were relatively stable.

### 3.3. The Low SC Volatility Variance Reflects the Significant Healing Effect of Landscape Colour on Subjects

In this paper, a visual analysis of the SC fluctuation variance in EDA (Figure 10) shows that the average SC fluctuation variance of 78 subjects is 5.594%, of which 59 subjects have SC fluctuation variance ≤ 6%, accounting for 75.641% of the total, and 48 subjects’ SC fluctuation variance ≤ 3%, accounting for 61.538% of the total; only 19 subjects’ SC fluctuation variances > 6%, accounting for 24.359% of the total. Additionally, according to PANAS of Figure 11 (PA = 43.31, NA = 17.95), the results show that after the modified wandering landscape immersive experience, the participants were emotionally stable, the psychological cognitive load was low, and the positive emotional arousal state was present. The results of the study show that after the immersive experience of the digital roaming landscape scene, the participants had low levels of positive emotional arousal and SC fluctuation variance, which confirmed that the participants’ stress level and psychological cognitive load were low, and the healing effect of the roaming landscape was significant, which is contrary to experimental Hypothesis 2: the lower the emotional arousal is, the more significant the healing effect. To further support the subjects’ evaluation of the healing effect of the seven types of landscape colours, according to the results of the Likert scale after the experiment (Figure 12), the participants rated the healing properties of landscape colours, from high to low, from Green > Yellow > Red > Blue > Cyan > Orange > Purple. The reliability and validity test of the scale results were performed by SPSS 21.0 software, and the results show that Cronbach’s coefficient (*α*) = 0.915, indicating that the reliability of the subjective questionnaire was reliable. Through the principal component analysis of the subjective questionnaire results, it was found that *KMO* = 0.884, which confirmed that the scale results were statistically significant (*p* < 0.01), indicating that the scale had good structural validity. In conclusion, the results of the study show that after the immersion experience, the SC fluctuation variance value is low, and the healing effect is significant. In the implementation of the landscape project, if the healing benefit landscape configuration is considered, then the green, yellow, red, and blue plant configuration can be selected first, and the cyan, orange, and purple as the main colour configuration of the healing landscape scene can be appropriately reduced.

## 4. Conclusions

According to the results of Section 3.1, the higher the fixation frequency is, the denser the colour concentration of the hot spot map is, which proves that the subjects are more interested in the roaming landscape and is consistent with the research results of [45]. The average variation trends of interest confirm that subjects have high interest in immersive experiences, which is consistent with the results of prior research results [46,47]. The results show that green, yellow, and blue plants have the highest concentration of interest and should be used as the main colour in landscape scenes. The relatively high concentration of interest in red plants indicates that the subjects are influenced by traditional Chinese culture, which is consistent with the results of [48]. In the implementation of landscape projects, the hue interval can be selected from the HSV colour space model in this paper, and the green plant colour can be selected from 0.3621 to 0.4625, the yellow from 0.1061 to 0.1374, the blue from 0.5508 to 0.6197, and the red from 0.9551 to 0.9930, although the above plant colours are used as the main configuration applications for increasing interest among populations of psychological subhealth populations. However, other landscape colours should also be appropriately matched to the landscapes to meet the actual needs of improving the ecosystem and enriching the visual space level of the landscape.

According to research result 3.2, the interest degree of subjects in landscape colours is correlated with positive emotional arousal. In this paper, the correlation analysis of the colour interest of seven types of landscapes and SC was analysed, and the results showed that blue, green, and yellow plants could cause positive emotional arousal in participants, and when watching the three colour plants, the participants had a small psychological cognitive load, calm mood, and small SC fluctuations. Blue space has a great promotion effect on human mental health, and blue scenes can arouse people’s positive emotions, which is consistent with the research results of [49,50]. Combined with the interview results after the experiment, 94.51% of the subjects indicated that their emotions tended to calm down when they saw blue plants, 85.71% indicated that their anxiety would be appropriately relieved and their body and mind relaxed when they saw green plants, and 83.52% expressed that they would have positive emotions such as positive sunshine when they saw yellow plants.

According to study Section 3.3, subjects in the digital roaming landscape scene had low emotional arousal, low SC fluctuation value, small psychological cognitive load, and small pressure value, and the healing effect of the digital roaming landscape scene was significant, which is consistent with the research results of [51] and [39]. The results of the control of the Likert scale show that the healing effect of green, yellow, red, and blue plants was the best, and the selection of EDA experiments and scales for comparison with the control group was because the ErgoLAB platform could only derive the SC values of the participants in the entire digital roaming landscape scene experience, so the combined measurement table obtained the healing effect evaluation of the seven types of landscape colours. The evaluation results were Green > Yellow > Red > Blue > Cyan > Orange > Purple.

Based on the preliminary field survey, middle of VR, EDA experiment, the period after the interview, and the Likert scale mode of the combination of subjective and objective research, the present experience—emotional response—of landscape healing is shown to have a logical relationship compared with the traditional expert interview method, and so, with the subjective evaluation study, population landscape colour preferences and healing effect research are more comprehensive, and subjective and objective indicators are mutually verified, which is consistent with the research results of [52]. In the preliminary questionnaire survey, people with mental subhealth associated with red, yellow, green, and blue plants had stronger positive emotions, which was consistent with experimental result 3; the combination of subjective evaluation and physiological indicators is more objective and scientific, which is consistent with experimental Hypothesis 3. In this paper, the data of subjects’ interest in landscape colour, mood change, and healing effect were obtained by combining subjective and objective methods, providing further reference and thinking for experimental research. The application value of the article is to heal the landscape design for actual construction enterprise design units, to provide some countries’ mental subhealth problems of anxiety and tension effective measures to reduce the mental subhealth state transition to body disease, mental illness, and even psychological crisis, and to provide new ideas for landscape healing effect evaluation.

This paper also has certain limitations. First, the proficiency of the handle operation as a control variable, the sense of vertigo, and the sensitivity of movement will affect the accuracy of the experimental data to a certain extent, and the pre-experiment can be used in the follow-up experiment. The subject perceives the experimental environment and the handle operation mode in advance, thereby improving the accuracy of the experimental data. Second, only 91 participants were randomly invited to participate in the experiment, and the overall sample size was still slightly insufficient. Affected by the normalisation of COVID-19, only the participants in the research area were invited in the experimental part, but the mental subhealth symptoms of the personnel in the research area were also more typical (the researchers were affected by the incident of the strangle, etc.), more representative, the experimental results were more scientific and reasonable, and the follow-up national and comprehensive research will be further deepened.

At the same time, the colour of the same landscape also changes with flowering and fruiting and the change in seasons [53,54,55]. This experimental study was selected to study the season of late spring and early summer, the month of April–May, also because the study area is richer in plant species at this time of the year, the landscape colour is more diverse, the results of the experiment in the green colour is more positive, and not because of the landscape appearing to be more green, but the results of the experimental data generated by the subjects compared to the analysis, and the study of landscape groups, communities, and artistic forms of recuperation also require further study. Finally, virtual eye-tracking and EDA experiments were conducted in this study, which only temporarily relieved the tension and anxiety of the subjects. In the future, we plan to conduct further research on the therapy cycle of patients with mental subhealth. In the context of the COVID-19 pandemic, this research shares the design, implementation, evaluation, acceptance, and human factors problems encountered in the application of digital medical technology in a wide range of public health environments, explores the effectiveness and added value of digital technology compared with traditional practices, and also investigates the practical exploration of users’ perception, recognition, and acceptance of digital medical technology.

## Figures and Tables

**Figure 1 ijerph-19-10986-f001:**
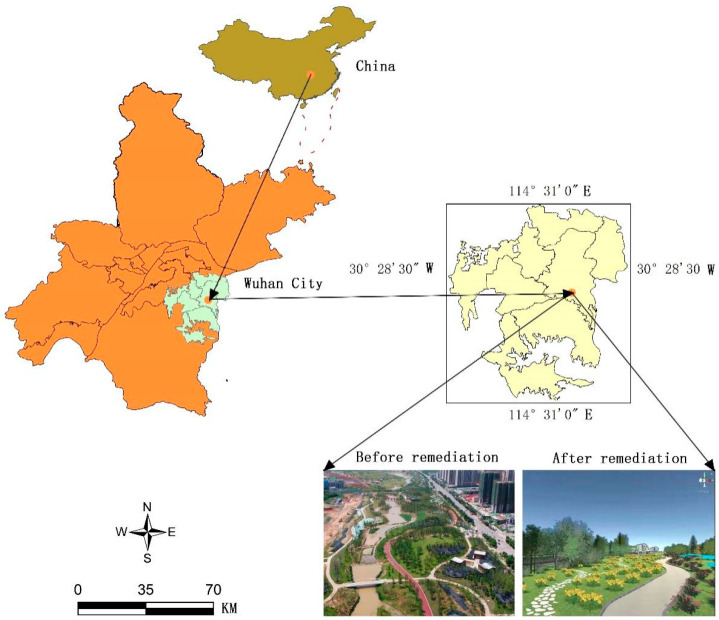
Location overview of study area in National High-Tech Zones. Source: Shot by DJI Inspire 2 (DJI Technology Co., Ltd, Shenzhen, China).

**Figure 2 ijerph-19-10986-f002:**
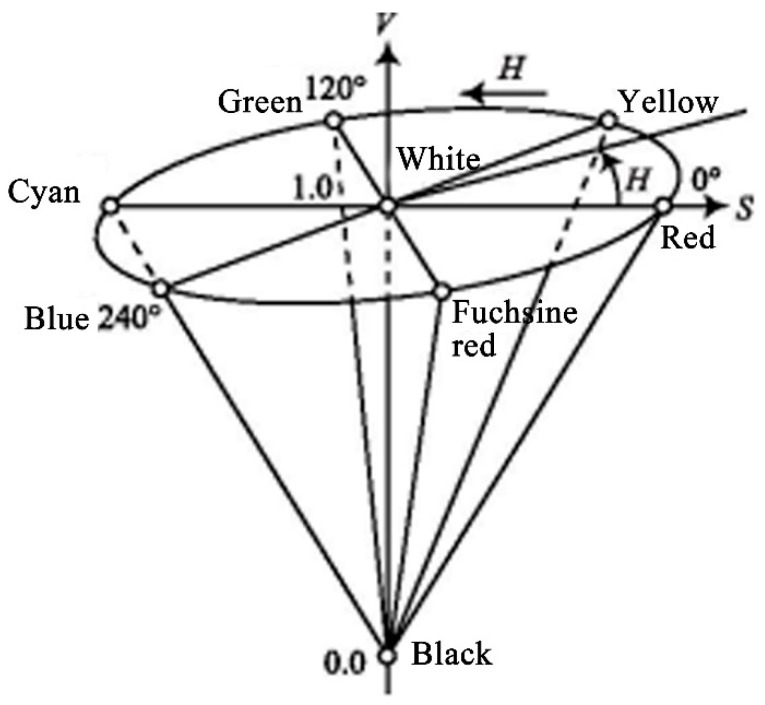
HSV colour space coordinate system.

**Figure 3 ijerph-19-10986-f003:**
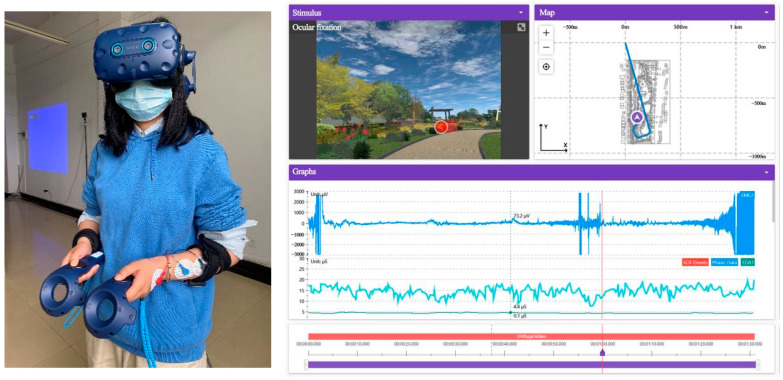
Subjects’ experiment scene and data monitoring interface.

**Figure 4 ijerph-19-10986-f004:**
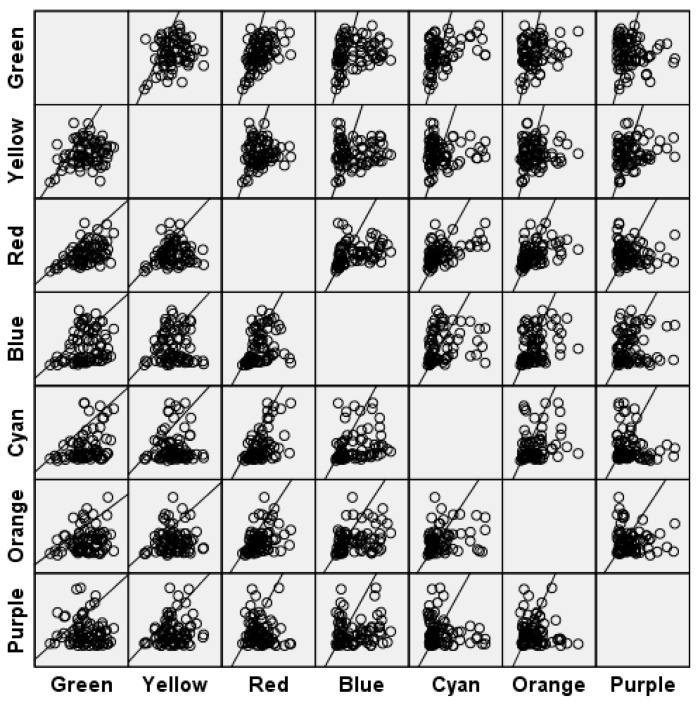
Influence factor of interest between colours scatter plot matrix. Source: Data exported from ErgoLAB program, SPSS software.

**Figure 5 ijerph-19-10986-f005:**
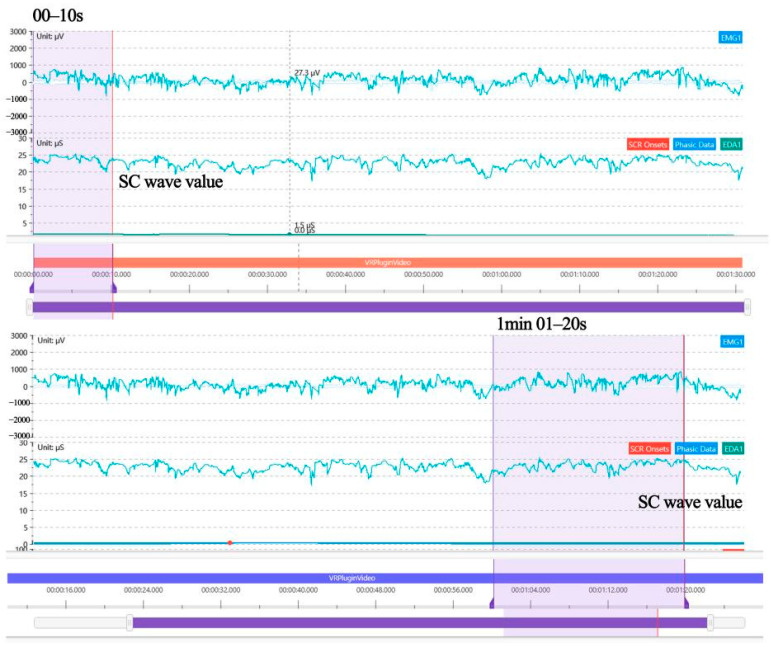
Extraction of SC fluctuation value. Source: Exported from ErgoLAB program.

**Figure 6 ijerph-19-10986-f006:**
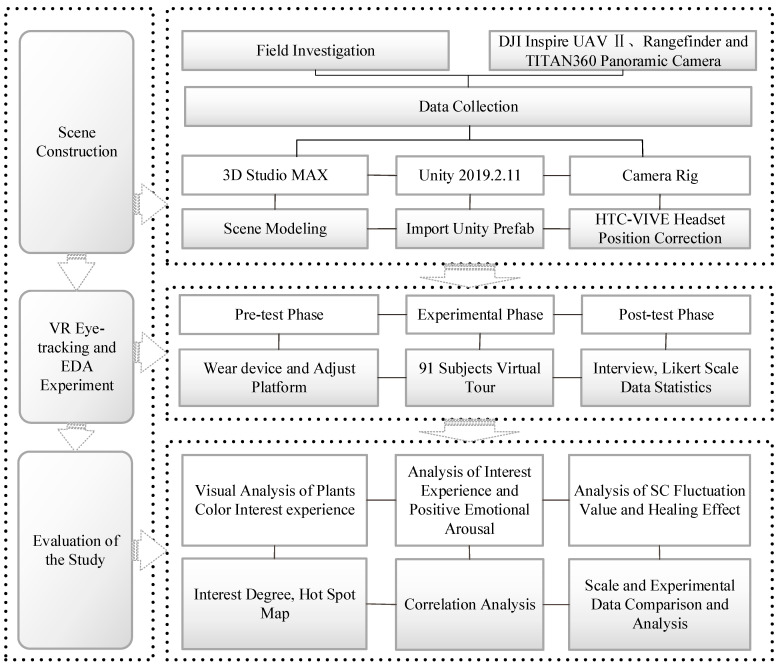
Reasoning diagram of landscape reconstruction design and healing effect.

**Figure 7 ijerph-19-10986-f007:**
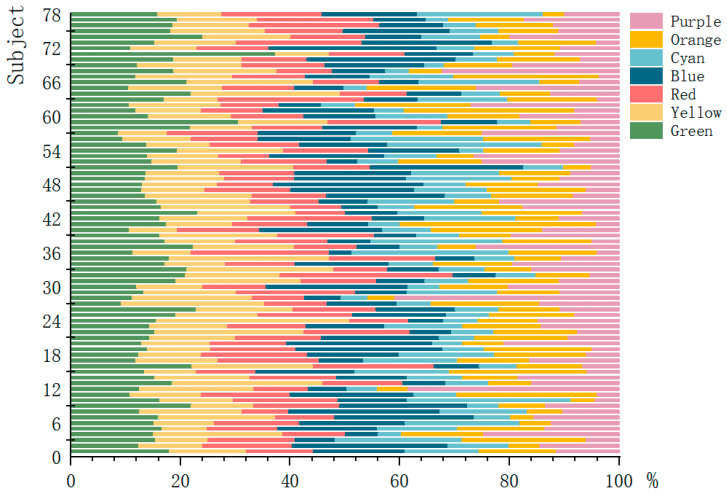
The changing trend of degree of interest of the 78 subjects. Source: ErgoLAB platform was used to export the data, and the overall interest of each subject in the immersive experience was classified as 100%. The figure reflects the proportion of interest of 78 subjects in seven types of landscape colours.

**Figure 8 ijerph-19-10986-f008:**
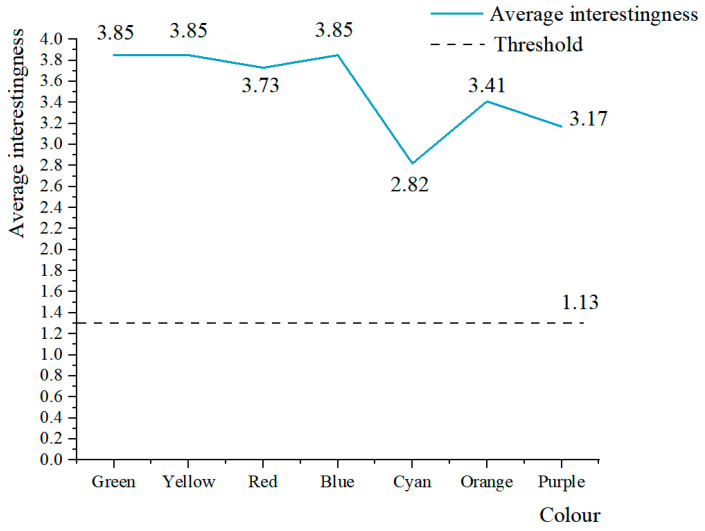
Overall mean variation trend of interest degree of 78 subjects. Source: ErgoLAB platform derived the data, and the weight threshold of level of interests was 1.13 when the duration of fixation was the longest. Due to the weak visual presentation of the variation trend of individual landscapes’ colour interest, the average trend chart of seven categories of landscapes’ colour interest of 78 subjects was made as an assistant.

**Figure 9 ijerph-19-10986-f009:**
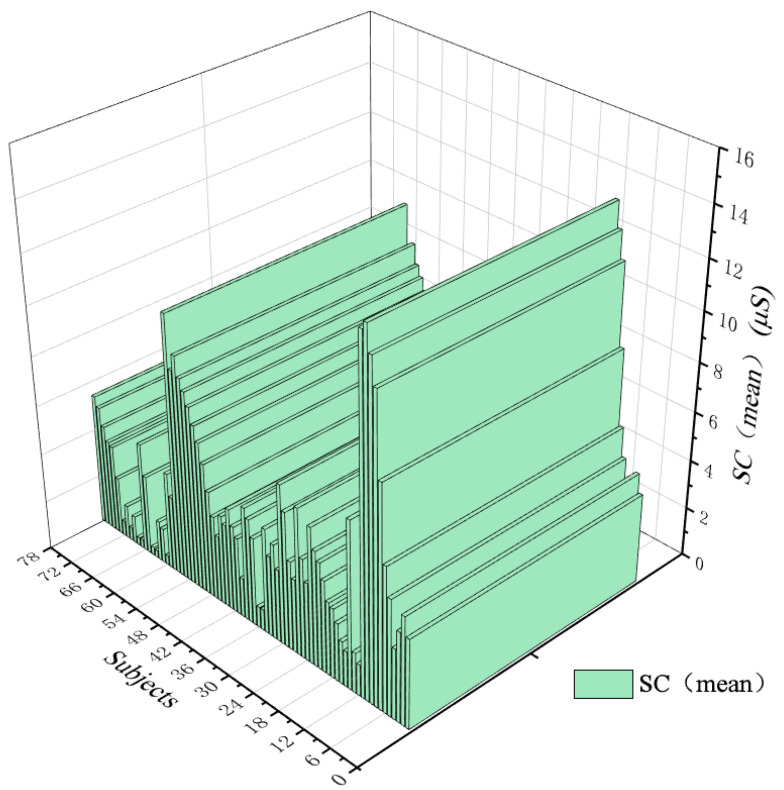
Trend chart of subjects’ SC average wave value change. Source: The ErgoLAB platform exports data, and the maximum SC average wave value is 14.27 μS, the minimum value is 0.29 μS, and the average value is 4.32 μS.

**Figure 10 ijerph-19-10986-f010:**
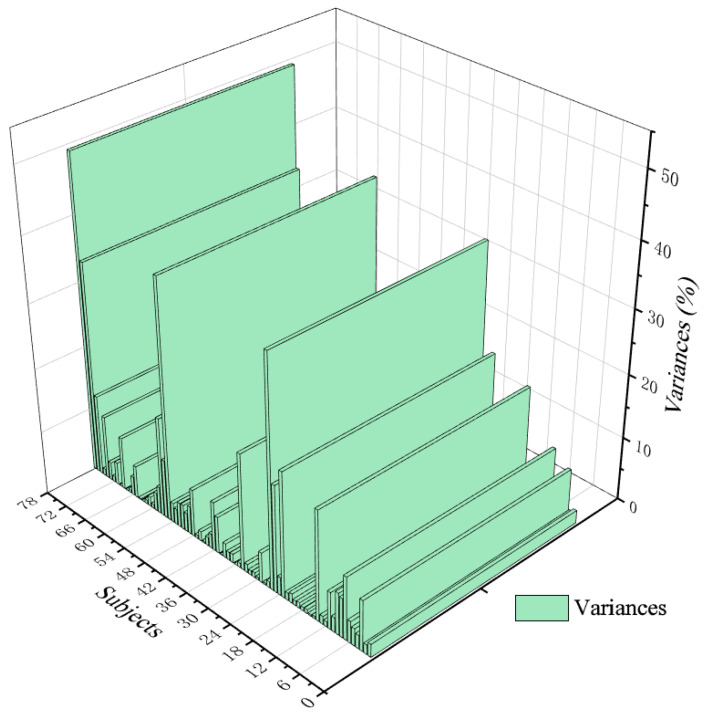
Variance trend plot of SC wave values. Source: ErgoLAB platform to export data, SC variance minimum 0, maximum value of 49%, average mean 5.594%.

**Figure 11 ijerph-19-10986-f011:**
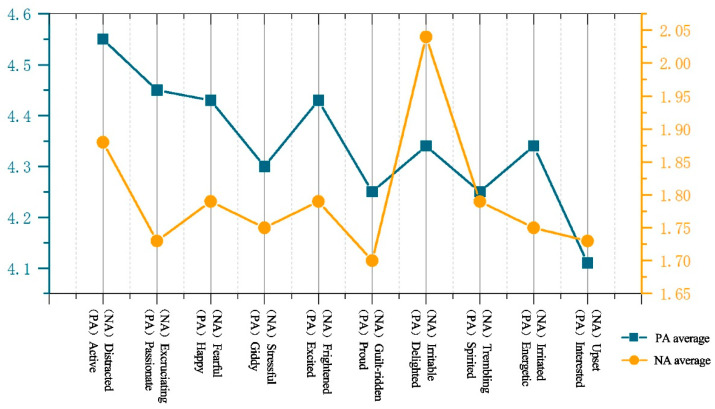
PANAS change trend chart. Source: After the experiment, the participants filled in the scale according to their own feelings of roaming the landscape experience, and the positive emotional adjective score ranged from 4.11 to 4.55 points, and the negative emotional adjective score ranged from 1.73 to 1.88 points.

**Figure 12 ijerph-19-10986-f012:**
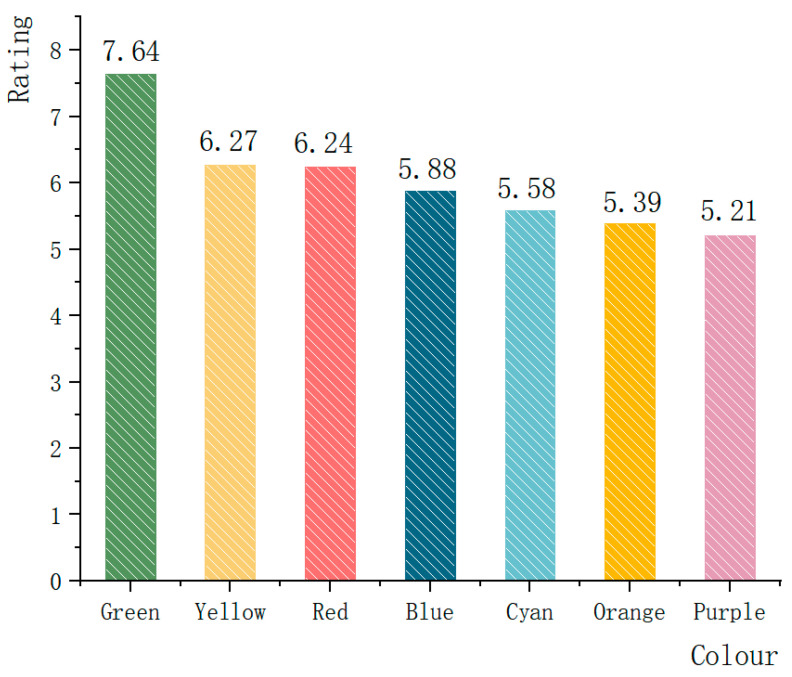
Likert scale healing rating trend chart. Source: Likert scale, colour healing rating of seven types of landscapes, and the figure shows the overall mean trend of subjects’ scores.

**Table 1 ijerph-19-10986-t001:** Results of the landscape colour positive and negative emotion adjective scales.

Category	Red	Orange	Yellow	Green	Cyan	Blue	Purple
Positive emotion score (PA)	34.77	28.32	32.03	32.45	27.67	32.21	28.3
Negative emotion score (NA)	21.64	22.65	20.73	18.89	23.19	21.43	23.87

**Table 2 ijerph-19-10986-t002:** H-tone component values of seven coloured plants.

Red	H	Orange	H	Yellow	H	Green	H	Cyan	H	Blue	H	Purple	H
Rose	0.9745	Tangerine	0.0409	Sunflower	0.1096	Camphor	0.4625	Indo calamus	0.5104	Forget-me-not	0.5636	Wisteria	0.6919
Peony	0.9551	Persimmon	0.0593	Yellow oak	0.1061	Chinese parasol	0.4571	Korean pine	0.5234	Dew grass	0.5508	Sea aster	0.7312
Chinese rose	0.9750	Orange	0.0697	Corn	0.1149	Cedar	0.3833	Bamboo	0.4896	Arctotis	0.5622	Lilac	0.7586
Red maple	0.9914	Pumpkin	0.0789	Gardenia	0.1145	Southern magnolia	0.3621	Equisetum hiemale	0.4915	Iris	0.6038	lavender	0.7181
Musa coccinea	0.9930	Day lily	0.0781	Gamboge	0.1269	Buxus	0.3925	Lotus leaf	0.4882	Iarkspur	0.6197	Radix gentianae	0.7473
Tomato	0.9929	Birch	0.0564	Rape flower	0.1374	Willow	0.4194	Wax gourd	0.5000	Hydrangea	0.6080	chinaberry	0.7033

Source: Combined with the result of HSV space obtained by RGB space conversion, the range of red plant colour selection in the H hue component is 0.9551~0.9930. Orange is 0.0409~0.0789; yellow is 0.1061~0.1374; green is 0.3621~0.4625; cyan is 0.4882~0.5234; blue is 0.5508~0.6197; purple is 0.6919~0.7586.

**Table 3 ijerph-19-10986-t003:** Statistical results of the correlation between interest and stress sentiment.

Interests	Sig	*p*	N
Green plants	0.019	<0.05	78
Yellow plants	0.026	<0.05	78
Blue plants	0.002	<0.05	78
Red plants	0.136	>0.05	78
Cyan plants	0.109	>0.05	78
Orange plants	0.725	>0.05	78
Purple plants	0.428	>0.05	78

## Data Availability

Not applicable.

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
