# Peer review of "Colour Preference and Healing in Digital Roaming Landscape: A Case Study of Mental Subhealth Populations"

_ijerph, 2022, doi:10.3390/ijerph191710986_

Round 1

Reviewer 1 Report

1.  This study uses virtual scenes, eye movements and other technical means to correlate visual and psychological preferences for landscape colour, which is a relatively novel research perspective.

2. There is a lack of connection between the title of this paper and the virtual digital roaming in the research content.

3. The lack of sufficient research review on the current status of the keywords in this paper in the introduction section does not reflect the significance and value of landscape colour research based on virtual scenes, and the relationship between heart subhealth and virtual scenes needs to be introduced more systematically.

4. Based on virtual experimental data from a research site in a high-tech zone in Wuhan, how can we improve the subhealth condition of most people's hearts nationwide.

5. In the experiment, the gender, age, occupation, income and working hours of the subjects are not described, which makes it impossible to judge the scientific validity and reasonableness of the experiment.

6. It is suggested that the photos of the virtual scenes and the overlay effect of the experimental data should be reflected in the paper.

7. The thesis only uses plants as a proxy for 'landscape colour', but in real scenarios, vegetation is only one part of the landscape composition, along with roads, buildings, sky and other colours, so a more diverse perspective is needed.

8. The findings suggest that there are differences in psychological healing between landscape colours, but there is a lack of research findings on how different colours are configured, or whether a higher psychological healing plant colour in a scene will necessarily improve people's psychological well-being?

Author Response

Point 1: This study uses virtual scenes, eye movements and other technical means to correlate visual and psychological preferences for landscape colour, which is a relatively novel research perspective.

Response 1: Thank you very much for your approval of our article. As you said, our research perspective is relatively new, so there is also a lot of room for improvement, and further exploration and refinement are needed in the future.

Point 2: There is a lack of connection between the title of this paper and the virtual digital roaming in the research content.

Response 2: Yes, we have revised the title according to your suggestion, and the revised title is Color Preference and Healing in Digital Roaming Landscapes: A Case Study of Mental Sub-Health Populations.

Point 3: The lack of sufficient research review on the current status of the keywords in this paper in the introduction section does not reflect the significance and value of landscape colour research based on virtual scenes, and the relationship between heart subhealth and virtual scenes needs to be introduced more systematically.

Response 3:These sections have been supplemented in the Introduction section Page 3, lines 103-112, and your valuable comments are appreciated.

Point 4: Based on virtual experimental data from a research site in a high-tech zone in Wuhan, how can we improve the subhealth condition of most people's hearts nationwide.

Response 4: 1. In the process of fieldwork, we also included the distribution of network questionnaires (questionnaires showed that patients with mental sub-health in 21 provinces and cities were filled out), at the same time, we interviewed psychologists in relevant hospitals in different provinces and cities through network video conferencing and telephone consultation, which also shows  the lack of rigor in our writing, section 2.2 Data Collection (Page 5, Lines 198-202,Page 5, Lines 220) in this paper has been supplemented.

2. The subjects in the study area were invited to the experimental part, also due to the influence of factors such as the prevention and control of the new crown epidemic, but the subjects in the study area were more typical (see specifically page 4, lines 160-165and annex A (Please see the attachment) for details), and also because we hope to apply the experimental results to the study area to observe the actual construction effects before conducting further studies and extensions in other areas, in Chapter 4 Section Conclusions and Discussion Page 18, lines 565-571, we also provide a future perspective on this.

Point 5: In the experiment, the gender, age, occupation, income and working hours of the subjects are not described, which makes it impossible to judge the scientific validity and reasonableness of the experiment.

Response 5: Thank you for your suggestion, the relevant contents have been placed in annex A (Please see the attachment), and the corresponding information has been added in the 2.4 VR eye-tracking experiment and EDA Experiment section Page 8, lines 279-284. However, we did not collect information on income, working hours, and other personal information during the experiment.

Point 6: It is suggested that the photos of the virtual scenes and the overlay effect of the experimental data should be reflected in the paper.

Response 6: 1. Since our scene is a landscape roaming experience, the selection of simple pictures in the text is easy to mislead, compared to the traditional two-dimensional flat pictures for experimental data acquisition, we are more immersive and realistic.

2. Because the interface of the experimental platform can only observe individual experiments, it is difficult to present the superposition effect of experimental data, and the forms of experimental data are relatively diverse and complex. The data visualization in the third result part of this paper has been iteratively optimized and can express the experimental data.

Point 7: The thesis only uses plants as a proxy for 'landscape colour', but in real scenarios, vegetation is only one part of the landscape composition, along with roads, buildings, sky and other colours, so a more diverse perspective is needed.

Response 7: In light of this valuable comment, we provide the following response. The immersive roaming scenes in this study include not only plant elements but also sky, clouds, roads and buildings (additional information is provided in Page 7, lines 257). We also included music in the roaming scenes for an immersive landscape color space experience through a combination of audio and visual.

Point 8: The findings suggest that there are differences in psychological healing between landscape colours, but there is a lack of research findings on how different colours are configured, or whether a higher psychological healing plant colour in a scene will necessarily improve people's psychological well-being?

Response 8: 1. At the end of each point of the results in Part III we give the results of the different color-related configurations. For example: page 13, lines 422-426; page 14, lines 444-450; pages 15-16, lines 482-486.

2. In the final conclusion and discussion section of the article, page 19, lines 579-582, we also show that this study temporarily alleviated the subjects' tension and anxiety states through virtual eye tracking and EDA experiments. In the future, we will conduct further studies on the treatment cycle of patients with psychological suboptimal health.

Reviewer 2 Report

The manuscript 'Landscape Colour Design and Healing Effect Based on Emotional Preference of Mental Sub-health' is an important and solid experimental study that provides relevant information and a new method for demonstrating the logical relationship between the digital landscape interests and experience-emotional awakening-healing effects and construction scheme for landscape colour configuration in the implementation of healing landscape projects. Especially the combination of subjective data and physiological parameters (mixed-methods design) is particularly positive.

However, I have a few suggestions to improve the manuscript:

  1. The English needs to be revised urgently.
    For example “Licht scale” on page 14, line 433.

  2. The references in the Conclusion and Discussion are incorrect. Here, for example, reference is made to the results in Chapter 4. These are, however, in Chap. 3. Please correct the references.

  3. The number of cases is quite small for statistical analyses with the large number of characteristics.

  4. Please specify why no validated instruments are used for the questionnaires?

  5. In chapter 2.5.3 refers to the analysis of emotional arousal and healing effects. While

    the emotional arousal is presented extensively. There are hardly any statements

    about the healing effects. Please specify the healing effects.

  6. Although the authors address the problem of the missing seasons in the

    discussion/limitations, the results can change significantly depending on the season. Plants and their associated colours are not static objects. They change colour throughout the season. Since the colour green in particular turned out to be particularly positive in the experiments, it is not more about the coherence of the landscape. Please specify in the discussion.

  7. Please classify the results even more closely in the overall public health discourse. What do the interesting and important results for the public health discourse in China and international tell us?

Author Response

Point 1: The English needs to be revised urgently.For example “Licht scale” on page 14, line 433.

Response 1: Thank you very much for such careful comments, we have proofread and revised the English in detail.

Point 2: The references in the Conclusion and Discussion are incorrect. Here, for example, reference is made to the results in Chapter 4. These are, however, in Chap. 3. Please correct the references.

Response 2: Your comments are very valuable, it is the result of our lack of rigor in our writing, and the questions you raised have been corrected in the fourth conclusion and discussion section.

Point 3: The number of cases is quite small for statistical analyses with the large number of characteristics.

Response 3: Thank you for your proposal, the subjects of this experiment are all mental sub-health patients, and 86.81% of the subjects indicated during the pre-experimental interview that emotional instability, tension and anxiety often arise in their life and work, and the experimental subjects' situation has been placed in annex A(Please see the attachment).

Point 4: Please specify why no validated instruments are used for the questionnaires?

Response 4: The questionnaires and scales of this study were analyzed for reliability and validity by SPSS software and are clearly indicated in the text, e.g., p. 5, lines 223-227, p. 15, lines 475-480.

Point 5: In chapter 2.5.3 refers to the analysis of emotional arousal and healing effects. While the emotional arousal is presented extensively. There are hardly any statements about the healing effects. Please specify the healing effects.

Response 5: Your suggestion is very important, the subjects of this study are mental sub-health patients and the ultimate goal is to perform the healing of relevant emotional aspects, such as anxiety and tension relief, indeed the arousal of emotions is extensive, therefore we did not give whether the arousal of positive or negative emotions, only the SC fluctuation variance values and the Likert scale were combined for the discourse of healing results. The data of the post-experimental positive and negative emotion adjective scale have been added in chapter 2.5.3, page 15, lines 462-466, and page 16, lines 490-494, and the results show that the subjects' negative emotions were effectively relieved, and combined with the post-experimental interview of the subjects showed that the healing effect was better after the roaming landscape scene experience, and the emotions such as anxiety and tension were relieved.

Point 6: Although the authors address the problem of the missing seasons in the discussion/limitations, the results can change significantly depending on the season. Plants and their associated colours are not static objects. They change colour throughout the season. Since the colour green in particular turned out to be particularly positive in the experiments, it is not more about the coherence of the landscape. Please specify in the discussion.

Response 6: Thank you for your valuable comments, which are specified in the discussion section, page 18, lines 573-578.

Point 7: Please classify the results even more closely in the overall public health discourse. What do the interesting and important results for the public health discourse in China and international tell us?

Response 7: Thank you for your valuable comments, which we have added in the Conclusions and Discussion section of Chapter 4, page 19, lines 582-588.

Reviewer 3 Report

The research tried to combine VR and dermatoelectric technology to analyze the healing effect of virtual landscape space. The research topic is novel and cutting-edge, but there are still some shortcomings to be improved. For example, the transformation relationship between the SU model and immersive experience in this paper needs to be further explained. How much healing effect these virtual Spaces have, and how to reflect their logical relationship?

Author Response

Thank you for your valuable comments and for your hard work and review of the article.

Response 1: First of all, the SU model is not mentioned in this article, the models of the roaming scenes were created by 3D Studio Max software, and the conversion from model to immersive experience is explained in chapter 2, section 2.3, page 5, lines 255-265 of the article.

Response 2: Your suggestion is very meaningful, as the subjects of this article are mental sub-health patients, and the ultimate goal of this study is to heal related emotions, such as anxiety, tension and other emotion relief.

Response 3: The article establishes the logical relationship of "degree of interest-emotional change-healing effect", which can be found in 2.6. Section, p. 11, lines 358-375.

Response 4: The specific logical relationships are briefly described as follows:

(1) The experimental data of the subjects were derived by constructing roaming scenarios in combination with field surveys and through virtual eye-movement and electrodermal experiments as well as questionnaire scales.

(2) The experimental results indicated that the subjects were in low emotional arousal in the scenario and the scenario was a low-stress environment.

(3) Emotional arousal was divided into positive and negative emotions, and the combination of experimental, questionnaire and scale data showed that it was verified that the subjects produced stronger positive emotions after the roaming experience, and the anxiety and tension were relieved.

Therefore, combined with the results of subjective questionnaire interviews and objective experiments, it was synthesized that the emotional level of the participants produced healing.
